



# Proposal of an instrument design to observe annual changes in Spectral Outgoing Radiation

Gerhard Schmidtke[1], Raimund Brunner [1], and Christoph Jacobi[2]

[1] Fraunhofer Institute for Physical Measurement Techniques IPM, Georges-Köhler-Allee 301, Germany

[2]Institute for Meteorology, Leipzig University, Stephanstr. 3, 04103 Leipzig, Germany

*Correspondence to*: Gerhard Schmidtke (gerhard_schmidtke@web.de)

**Abstract:** From the wide range of possibilities, we propose an instrument capable of measuring annual changes in global Spectral Outgoing Radiation (SORa) from the entire Earth's surface between 200 nm and 1100 nm with a stability of 0.1

$Wm^{-2}$ over a period of one solar cycle or beyond. Photomultiplier tubes (PMTs) as detectors provide data with a cadence of one second and high dynamic range. Based on Total Solar Radiation TSI(t) data with a stability of 0.01 $Wm^{-2}$ per year, Spectral Solar Irradiance SSI(t) can be derived and normalized to ΣSSI(t)=TSI(t) for using the Sun as a referenced radiation source supported by solar modeling. Calibrated by SSI(t), a set of 12 spectrometers with 60 PMTs in total and 16 photometers simultaneously detect SOR(t). This database can also be provided to calibrate other space instruments to allow

improved comparison of results. In previous missions in space, it has already been shown that the spectrometer design can detect both solar and terrestrial radiation with high dynamic range. The established measurement technique compensates for degradation through repeated calibration. The instrument also enables the determination of the global green Earth coverage and its annual changes by measuring chlorophyll absorption from 350 nm to 490 nm and 620 nm to 690 nm and green backscatter from 500 nm to 600 nm. Mapping the Earth will also make it possible to track annual local changes in green

coverage and to assess the impact of different climate policies and climate engineering actions. Another aspect is the derivation of a correction parameter for the Earth Energy Imbalance derived from changes in green areas. Data evaluation can also include determining further parameters such as the Normalized Difference Vegetation Index, the Enhanced Vegetation Index, and the Global Leaf Area Index.

*Key words.* Spectral Outgoing Radiation, Solar Spectral Irradiance, remote sensing, calibration, Earth Energy Imbalance, degradation, global green coverage

## 1 Introduction

A key parameter for observing changes in the Earth´s energy budget is the global Spectral Outgoing Radiation (SOR). We propose an instrument to measure annual changes in SOR (SOR$_a$) over a solar cycle or beyond, based on the method of the



Solar Auto-Calibrating Extreme UltraViolet Spectrometers SolACES (Schmidtke et al., 2014; Schäfer et al., 2017). The versatile SolACES measurement method is based on an aperture wheel with 48 measurement windows for different optical components or subsystems. The proposed instrument is equipped with simultaneously measuring 12 spectrometers and 16 photometers. 20 radiation attenuators enable the adjustment of the Solar Spectral Irradiance SSI(t) to natural SOR values, the calibration of SOR recording devices and the checking of the linearity of their detectability. With this equipment, $SOR_a$ can

be measured in the spectral range from 200 nm to 1100 nm with sensitive photomultiplier tubes PMTs as detectors in the spectrometers and photometers with an expected stability of 0.1 $Wm^{-2}$ to be achieved through repeated calibration. We call the instrument Spectral Outgoing Radiation Auto-Calibrating Spectrometers SORACES because the measurement method and basic geometry are identical to those of SolACES.

One aspect is the determination of plant-covered area of the Earth and its annual changes. Increasing levels of carbon dioxide

in the atmosphere and rising temperatures may increase the green area, while desertification reduces it. Measuring the global area covered by green Earth would also allow local resolution to map the Earth and track annual local change, e.g. to check the effectiveness of various climate policy measures such as climate engineering actions. A possible increase in green area due to global warming leads to an increase in biomass, which stores carbon from atmospheric carbon dioxide and solar energy (Piao et al., 2019), which promotes cooling of the Earth. In the process of absorbing and storing solar energy, the

amount of Total Outgoing Radiation (TOR) decreases, causing an apparent increase in the annual mean Earth Energy Imbalance (EEI). It is the objective to determine changes in the global and local green SOR ($SOR_{agreen}$) from year to year, which we call annual $\Delta SOR_{agreen}$ with backscattered solar green radiation in the spectral range from 500 nm to 600 nm. Reflected and backscattered green solar radiation from a variety of different types of grassland, agricultural crops and forests is recorded to be <25 % (Virtanen et al., 2020). One contribution to stop global warming in the long term is to heavily store

solar energy in biomass while reducing atmospheric carbon dioxide concentrations.

The determination of global data sets in the spectral range from 350 nm to 690 nm is to distinguish between absorption of Spectral Solar Irradiance (SSI) by chlorophyll and emission from green plants. It is also possible to record chlorophyll fluorescence in oceans and other bodies of water.

Besides using a different measurement method, one reason for doing this is to correct for the annual changes in Earth Energy

Imbalance ($\Delta EEI_a$) by quantifying global green areas annually, which will provide $\Delta SOR_{agreen}$.

Our proposed method is integrative in that $SOR(\lambda)$ would be derived from annual mean measurement data. It removes altitude change artifacts, footprint-based Field of View (FOV) estimates, degradation and long-term instrument drift by deriving annual mean values $SOR(\lambda)_a$ from repeatedly calibrated instruments. Calibration is performed by using the Sun as a referenced radiation source. Cloudiness, seasonal changes in radiative reflectance, angle of incidence from Total Solar

Irradiance (TSI) and other factors affecting the mean annual $SOR_a$ data are averaged annually in the same way. A separate determination of these parameters would lead to significantly greater inaccuracies than 0.1 $Wm^{-2}$. Careful calibration and long-term stability are prerequisites for providing accurate data.



The method requires the Earth to be observed with the same measurement sequence from year to year. Also the data analysis should be repeated year after year. Worldwide coverage of the observables is to be achieved several times a year.

Data evaluation would involve applying the Normalized Difference Vegetation Index method (NDVI, Tucker and Sellers, 1986), the Enhanced Vegetation Index (EVI, Huete et al., 2002), and the Global Leaf Area Index (LAI, Fang et al., 2019). It is of special interest to compare results derived from different techniques of measurements and long-term calibration in space with repeatedly calibrated instruments.

The SORACES instrumentation, calibration and data analysis are explained in the following sections. Section 2 describes the

instrument and measurements. In Section 3, the calibration in both laboratory and space is explained in detail, while Section 4 describes orbit and spacecraft operation aspects. Section 5 concludes the paper.

## 2 Instruments and measurements

The schematic representation of SORACES (see Fig. 1) refers to Solar Auto-Calibrating EUV Spectrometers SolACES (EUV – Extreme Ultraviolet) with a double-circular wheel of 48 optical apertures. This instrument was operated on the

International Space Station for a period of 9 years (Schmidtke et al, 2014; Schäfer et al., 2017). The absolute calibration in space was performed using ionization chambers as absolute detectors and was based on the absorption of EUV photons by noble gas atoms creating ion-electron pairs. The conversion of the ion current into the number of ions gave the number of photons and thus the EUV Spectral Solar Irradiance SSI(t).

The calibration of SORACES requires knowledge of the correct SSI(t) in the wavelength range from 200 nm to 1100 nm,

which should be available from Solar Auto-Calibrating XUV/IR Spectrometers SOLACER (Schmidtke et al., 2019) with the wavelength range from the EUV spectral region to 3000 nm supported by solar modeling to also include the wavelength range >3000 nm. The extension of the EUV wavelength range from SolACES to the infrared in SOLACER is achieved by using Digital Absolute RAdiometers (DARA, Schmutz et al., 2013) or Active Cavity Radiometer Irradiance Monitor (ACRIM, Scafetta and Willson, 2014) as absolute radiation detectors. These devices provide TSI(t) data with a stability of

0.01 Wm$^{-2}$ per year (Ball et al., 2016; Finsterle et al., 2014). Normalizing TSI(t) to ΣSSI(t) adjusts the stability of both quantities so that in-space spectrometers and photometers can be calibrated with high stability to compensate for the instrument degradation.

The SOLACER method for deriving SSI(t) uses referenced TSI(t) data (Schmidtke et al., 2019). By splitting the TSI into SSI(Δλ) sub-ranges by narrow and medium band filters at the entrance aperture of planar grating spectrometers with their

high radiation throughput, each sub-range SSI(Δλ) is determined by a TSI instrument in terms of Wm$^{-2}$. Since Δλ ranges cover the XUV-IR spectral domains up to 3000 nm, the planar grating spectrometers are calibrated from the EUV through the NIR spectral range. By modeling the Sun, an extrapolation of the spectral range >3000 nm could be achieved. Although the relative spectral distribution of SSI(t) is not perfectly represented due to ongoing changes in solar activity, the correct





energy content of the entire SSI satisfies the requirements for climate modeling as well as for using SSI(t) to calibrate other

instruments in space, which is the main advantage compared to other methods.

The use of bandpass filters also enables the precise determination of the contributions of higher optical orders as well as the

contribution of scattered light to the recorded signals of the spectrometers (Schmidtke et al., 2014).

Instrument calibration in space based on these SSI(t) data is a key issue.

The calibration of SOR recording instruments has to adjust SSI(t) to natural SOR values with about 5 orders of magnitude

difference in radiation by attenuators and other tools. Linearity checks of the detectivity also has to be accomplished through

attenuators.

Since the solar variability of SSI(t) increases with decreasing wavelength, knowledge of this data during both solar quiet and

active periods is necessary in climate modeling to calculate its non-negligible contribution to the warming of the Earth's

middle and lower atmosphere and to the Earth's surfaces (Matthes et al., 2017).

**2.1 SOR measurements with spectrometers**

The main process of absorbing solar energy and converting it into bio-mass takes place in the chlorophyll green of the plants.

We aim to quantify this parameter by measuring $SOR_a$ and its changes in the spectral ranges relevant for chlorophyll. Of the

six types of chlorophyll, chlorophyll *a* and chlorophyll *b* are the most common ones (Milne et al., 2015). Type *a* absorbs

mainly from 350 nm to 450 nm and 650 nm to 690 nm wavelength, while type *b* absorbs from 400 nm to 490 nm and 620

nm to 650 nm. Since green light is reflected from 500 nm to 600 nm (Fig. 2), a spectral resolution of 5 nm would be

sufficient to observe plant coverage. Absorption by chlorophyll <350 nm and >690 nm are of lesser interest, although

included in the measurements down to 200 nm and up to 1100 nm.

In the spectral range of 200 to 1100 nm, the SSI(λ,t) also interacts with aerosols, trace gases including ozone and clouds

resulting in partial absorption and scattering in the atmosphere.

Since biomass originates from forests, steppes and grasslands with a large biodiversity with different degrees of absorption,

no applicable specific absorption coefficient is known. Although reflectivity of green light from leaves exceeds 20 %

(Virtanen et al., 22020), we calculate the signal-to-noise ratios of the spectrometers for a reflectivity of 10 % in the spectral

range from 200 nm to 800 nm.

The compact Rowland spectrometers of the ASSI on board the San Marco 5 satellite mission (Schmidtke et al., 1985a) are

well suited for SORACES instruments in terms of size, weight, power and performance (Fig. 3). The spectrometers

demonstrated the ability to record solar and Earth radiation with the same instrument. For example, toroidal optical gratings

with a radius R=115.5 mm and d=333, 500, 1000, and 2500 lines per mm could extend the spectral range from 120 to 3000

nm. SORACES with 48 apertures reduces the spectral range to <1100 nm for optimal detection of green and chlorophyll

ranges and is equipped with 12 ASSI-type spectrometers, 16 photometers and 20 radiation attenuators. Most detectors of

spectrometers and photometers are Hamamatsu R13096 PMTs weighing of 45 g. Each spectrometer contains 5 PMTs, which



detect radiation at spectral intervals from 5 nm to determine absorption by chlorophyll and intervals from 10 nm to 40 nm to measure green emissions and other contributions to TOR, e.g. to discern clouds.

The components of the spectrometers are fixed. In the double spectrometers, the angles of the detectors to the grating are set in such a way that the gaps in the wavelength intervals between the detectors complement each other. The widths of the exit
slits are either adjusted to the detector areas in order to expand the wavelength interval of each detector and to cover broad spectral intervals, or narrowed to increase spectral resolution. The latter should be highest from 350 nm to 690 nm, adjusted for chlorophyll absorbance.

Since the Earth is not a perfect sphere, we propose to use the following parameters to derive the expected maximum radiation flux $\Phi(\lambda,\Delta\lambda,t)$ to observe global changes in the SOR data: the Earth's radius is R=6378 km and TSI /4=1361/4
$Wm^{-2}$. $\Phi$ is given in units of photons $m^{-2}s^{-1}$, which is equivalent to $Wm^{-2}$.

For Hamamatsu PMTs R1309, the radiation flux $\Phi_p(\lambda,\Delta\lambda,t)$ recorded with Airglow Solar Spectrometer Instrument ASSI spectrometers is given in photons $m^{-}s^{-1}$, assuming that SOR(t)=60 $Wm^{-2}$ (Wild et al., 2013) in the spectral range from 200 nm to 800 nm (Table 1). SOR includes reflected, diffuse, and scattered outgoing solar radiation.

Knowing $\Sigma SSI(\lambda,t)=TSI(t)=I$, we can convert $SSI(\lambda,t)$ from $Wm^{-2}$ to the equivalent $\Phi$ given in photons ph $m^{-2}s^{-1}$. Radiance
would amount to I /($4\pi$) in ph $m^{-2}s^{-1}sr^{-1}$. For example: 1 R (in Rayleighs) = $10^{10}$ ph $m^{-2}s^{-1}$ =$10^{10}/(4\pi)$ ph $m^{-2}s^{-1}sr^{-1}$.

The ASSI FOV ($FOV_{ASSI}$) is 15 degrees in the plane of the grating and 30 degrees in the plane of the entrance slit. From a satellite at an altitude of 800 km, the spectrometers observe a terrestrial area of $FOV_{ASSI}\approx87362$ $km^2$ out of $5.10*10^8$ $km^2$ at the Earth's surface $E_s$,

$$E_s /FOV_{ASSI}\approx5.84*10^3= Q_{ASSI.} \qquad (1)$$

The factor $Q_{ASSI}$ relates the Earth' surface to $FOV_{ASSI}$.

With the width of the entrance slit of 0.3 mm and a length of 10 mm, the area $A_{slit}$ of the slit becomes $A_{slit}=0,03*10^{-4}$ $m^2$. A grating of 1000 lines/mm is chosen with the spectral reflectivity of $R_G=0.05$. The angle of the entrance slit is 30 degrees, and the detectors are arranged at angles from +25 to -23 degrees, which cover the spectral range from 109 nm to 922 nm. The spectral resolution is $\Delta\lambda_{sp}=6.84$ nm. The efficiency of the photomultiplier tube is $\eta_G(\lambda)$. The signals $S(\lambda,\Delta\lambda_{sp})$ of the
spectrometers in counts per second (cps) with clouds are

$$S_{CL}(\lambda,\Delta\lambda_{sp})=\Phi(\lambda,\Delta\lambda)*\eta_G(\lambda)*F_{slit}*R_G*\Delta\lambda_{sp}/Q_{ASSI}, \qquad (2)$$

with the constant factor CF,

$$CF=F_{slit}*R_G*\Delta\lambda_{sp}/Q_{ASSI} =1.76*10^{-10}. \qquad (3)$$

According to Wild et al. (2013) the signals without clouds amount to $S(\lambda, \Delta\lambda_{sp})$ in relation to those with clouds,
$S_{CL}(\lambda\Delta\lambda_{sp})=0.24*S(\lambda,\Delta\lambda_{sp})$.

Table 1 summarizes the expected signals of $SOR(\lambda,\Delta\lambda)$ from 200 nm to 800 nm according to the ASSI instrument aboard the San Marco 5 satellite for $S_{CL}(\lambda,\Delta\lambda_{sp})$ and $S(\lambda,\Delta\lambda_{sp})$.



The stability of the PMT data depends on the dark current of 20 nA, as stated for the Hamamatsu PMT: A gain of $1:10^7$ and a transit time of the pulse of 22 ns lead to counts per second cps $>10^6$ and the signal $S_{ASSI}=1.6*10^{-6}$ A. With a dark current of 20 nA, the statistical uncertainty results in $U=20*10^{-9}/1.6*10^{-6}=0.0126$ s$^{-1}$. If the measurement is repeated for 1 minute, the statistical uncertainty is $U_{minute}=0.0126/\sqrt{59}=1.6*10^{-3}$ per minute and $U_{orbit}=2.4*10^{-4}$ per sunlit period of ≈45 minutes of a satellite orbit. Then there should be accurate data to categorize and investigate selected geographic areas for SOR and for SOR changes.

**Table 1: Column 1: wavelength in λ; column 2: SOR(λ,Δλ) at λ with Δλ = 1 nm and SOR( $λ_1$ to $λ_2$,Δλ) = (SOR($λ_1$,Δλ) + SOR($λ_2$,Δλ))\*100/2 with ΣSOR (800 to 700 + 700 to 600 + 600 to……200 nm) = 60 Wm$^{-2}$; column 3: energy $E_{ph}(λ)$ of a photon; column 4: PMT quantum efficiency $η_G(λ)$; column 5: photon flux $Φ(λ, Δλ)* η_G(λ)$; column 6: expected count rates per second $S_{CL}(λ,Δλ_{sp})$ over clouds at spectral resolution of 6.84 nm; column 7: expected cps $S(λ,Δλ_{sp})$ without clouds at spectral resolution of 6.84 nm.**

| λ<br>nm | SOR(λ,Δλ)<br>Wm$^{-2}$ nm$^{-1}$ | $E_{ph}(λ)$<br>Joule | $η_G(λ)$<br>% | $Φ(λ,Δλ)*η_G(λ)$<br>ph m$^{-2}$ s$^{-1}$ nm$^{-1}$ | $S_{CL}(λ,Δλ_{sp})=Φ(λ,Δλ)*η_G(λ)*CF$<br>counts per second cps | $S(λ,Δλ_{sp})$<br>cps |
|---|---|---|---|---|---|---|
| 800 | 0.096 | $0.250*10^{-18}$ | 7 | $2.69*10^{16}$ | $4.74*10^6$ | $1.2*10^6$ |
| 700 | 0.112 | $0.286*10^{-18}$ | 14 | $5.47*10^{16}$ | $0.97*10^7$ | $2.3*10^6$ |
| 600 | 0.136 | $0.333*10^{-18}$ | 18 | $7.34*10^{16}$ | $1.29*10^7$ | $3.1*10^6$ |
| 500 | 0.144 | $0.400*10^{-18}$ | 22 | $7.92*10^{16}$ | $1.40*10^7$ | $3.4*10^6$ |
| 400 | 0.112 | $0.500*10^{-18}$ | 27 | $6.05*10^{16}$ | $1.07*10^7$ | $2.6*10^6$ |
| 300 | 0.048 | $0.667*10^{-18}$ | 29 | $2.09*10^{16}$ | $3.78*10^6$ | $0.9*10^6$ |
| 200 | 0.001 | $1.000*10^{-18}$ | 21 | $2.10*10^{14}$ | $3.70*10^4$ | $0.9*10^4$ |

The spectral resolution should be adjusted to the requirements of the spectral regions of the observables.

With 15 orbits per day, and 300 days out of 365 a year, there is a large number *N* of repeated measurements to improve the stability of the data using $N^{-1/2}$ by cumulating the measurements and averaging to one mean data point per year. Similarly, all of the annual $TSI_a$, $SSI_a$ and $SOR_a$ data are based on a large number of measurements with the largest number *N* with a cadence of 1 s.

**2.2 SOR measurements with photometers**

16 photometers are PMTs equipped with bandpass filters from 5 nm to 40 nm wide. ASSI type stray light baffles form a FOV of 10$^o$, which corresponds to ~140 km on the ground. Since the minimum effective area of the Hamamatsu PMT is 10\*24 mm$^2$, and there is no photon loss at an optical grating, both parameters correspond to the increased sensitivity of a factor of 1600 compared to spectrometers. Therefore, there is a great variety of options for selecting FOV and wavelength





intervals for photometers to observe $\Phi(\lambda, \Delta\lambda)$ in the spectral range from 200 nm to 1100 nm. High sensitivity of photometers is demonstrated by rocket missions recording auroral emissions at levels of noise as low as ≈10 Rayleigh at 557 nm (Schmidtke et al., 1985b). This sensitivity enables very high local resolution. These type of instruments must even be
attenuated for using a 10° FOV.

## 3 Instrument calibration

The new method compensates for instrument degradation through repeated in-space calibration based on referenced TSI(t) data. Changes in TSI(t) cause corresponding changes in SSI(t) and SOR(t).

Given the wide energy ranges of TSI and SSI versus SOR radiant flux by about 5 orders of magnitude, calibrating
instruments that overcome this difference is a challenge. Although radiation sources with energy levels close to TOR are available in the laboratory, this requirement is not met to calibrate instruments at the SSI level with a similar spectral composition from EUV to near-infrared. Therefore, calibration in the laboratory differs from that in space. Both calibration methods are explained separately in the following.

### 3.1 Instrument calibration in the laboratory

The calibration in the laboratory is considered provisional due to the incomprehensible degradation (Schmidtke, 2015) during the various activities after removing the instrument from the vacuum vessel. Integration into the payload and various tests, transport to the launch site and rocket tests, launch and commissioning activities change the efficiency of optical components in such a way that no further changes can be predicted or inferred during the mission in the harsh space environment. Therefore, laboratory calibration serves for adjusting the operational parameters of the spectrometers and
photometers to TOR levels and to track calibration changes during ground operations, testing and qualification of the SORACES instrument.

### 3.2 Instrument calibration in space

During the mission, bipolar water molecules detach from surfaces and change the performance of the components in the ultra-high vacuum at pressures below $10^{-6}$ hPa, which is not realized in the laboratory. Outgassing from drill holes with
screws, pores, insolating wires and circuit boards of the instruments contain hydrocarbons which crack in space and deposit on surfaces such as coatings. Interaction with atomic oxygen is another source of uncertainty.

Using the Moon as a source of radiation for calibrating instruments in space brings major limitations in terms of accuracy: The Moon is a secondary source of radiation that requires the precise knowledge of the primary source TSI(t), which varies with solar distance to the moon and with solar activity. The reflection of sunlight from the amorphous lunar surface also
depends on wavelength and angle of incidence, so the moonlight is not linear with TSI(t) over the spectral range. The



difficult assessment of the lunar surface also complicates spectral measurements (Wu et al., 2018). Determining the exact distance from the Moon and the actual angle of incidence from the Sun over the Moon to the spectrometers is also difficult. In space, we propose to use the SOLACER instrument to determine the SSI(t) and level it to SOR for wavelengths <1100nm by attenuators and other tools applying a combination of the following space-proven methods including linearity checks:

1. Although PMTs provide signals up to 7 orders of magnitude dynamic range, the linearity is between four and five orders of magnitude.

  2. Up to five orders of magnitude attenuation is obtainable by drilling ten 10 μm holes in a 1 cm diameter thin metal plate. Holes and slits have a linear radiant throughput for every wavelength. Atomic oxygen causes no changes.

    3. Two orders of sensitivity changes were realized with 10 preamplifier sensitivity levels (Schmidtke et al., 1974).

4. Changes of the high PMT voltage provide another tool to increase the dynamic range.

    5. Diffraction filters provide up to seven orders of magnitude attenuation (Schmidtke, 1968, 1970; Schmidtke et al., 1974; Nishimoto et al., 2021).

An attenuation of 5 orders of magnitude is available for the calibration of ASSI spectrometers with SSI(t), e.g. by Item #2 or by combining Item #2 with Item #5. The degree of attenuation could be further increased. Linearity should be verified by using Item #2 with transmissions of $10^{-1}$, $10^{-2}$, $10^{-3}$, $10^{-4}$ and $10^{-5}$. With the knowledge of the metal temperature and the coefficient of expansion, the changes in attenuation, if any, should be well known. Scanning the Sun along two axes will also provide the simultaneous response of the spectrometers and photometers as a function of the Sun position in their FOV.

Since our method is based on the continuous determination of TSI(t), international cooperation on the accuracy of this data has a high priority. Therefore, the establishment of a TSI(t) index analogous to the Solar Radio Flux Index F10.7 and the Zurich Sunspot Number should be aimed for in the future as a common reference of all TSI-related work. Then the Sun could be used as a common reference source for SSI-related work as well.

**4 Orbit and operations in orbit**

One parameter of concern is the spacecraft's decreasing altitude during the mission due to atmospheric drag which also varies over a solar cycle. For example, the Earth Radiation Budget Satellite (ERBS) was placed in a processing orbit with a 57° inclination and an altitude of 611 km, falling to about 590 km (Wong et al., 2018, their Fig. 8, upper frame). While the orbital altitude decreased almost constantly during low solar activity, it changed significantly due to increasing drag between 1989-1993 during maximum activity of solar cycle 22 and the starting maximum activity of solar cycle 23 in 1999-2000. From an altitude of 610 km a SOR instrument with a FOV of 10° observes an area of 8943 $km^2$ on ground and from an altitude of 590 km it is 8366 $km^2$, therefore the SOR differs by 7 %. A homogeneous SOR is not expected from both areas, so a correction for altitude changes remains uncertain.

To reduce the influence of drag on the accuracy of the data, an orbital altitude of 800 km is chosen, with the satellite drag being more than an order of magnitude lower than the drag of a satellite launched at an orbital altitude of 610 km. At about



800 km altitude, the altitude is stable over a period of 11 years. Since the outgoing radiation depends on local time, a Sun-synchronous orbit is chosen. This was clearly suggested by the results from the PICARD mission (Ball et al., 2016). The

same local times on the sunny and shady sides simplify the data evaluation and the local assignment of the SOR data with GPS coordinates. This is intended to track annual changes with calibration-like stability. With a $FOV_{ASSI}$ of $30^o$ x $15^o$, the entire area at the equator is observed approximately every 6th day and up to 15 times a day in the polar regions. Based on 300 measuring days per year, 47 complete data sets are available to derive changes in annual $SOR_a$ data. With SORACES pointing toward the center of the Earth and a cadence of 1 s, 1.3*10^7 data points lead to the mean $SOR_a$ data at the day side

of the orbits. The daily sequence of measurements must follow the same schedule for 300 days each year.

A FOV of 10 degrees of photometers corresponds to a diameter of 170 km on the Earth's surface. One orbit takes 100 minutes, which is 25 seconds to observe a circular area of 170 km in diameter. In order to derive categorized or scene-type local $SOR_a(\lambda,\Delta\lambda)$ data from oceanic, urban, desert, glacier, agricultural, mountain, and other areas, up to 46 complete data sets per year could be recorded at the equator, with the number increasing significantly towards the poles. In this way, a map

of the Earth could be created to track local changes in general, as well as to support climate policies. The local spatial resolution on the Earth's surface can be further increased.

Local energy accounting would be a major challenge to derive energy input and outgoing radiation to quantify the part of energy either stored in forests or other areas, or released during energy use or forest fires. The evaluation of the spectral composition <1100 nm should support activities to answer this question.

A satellite mission with the Earth-oriented instruments on board should measure changes in SOR over a full Solar Cycle. With this rigorous operational plan and using a known solar radiation source for calibration, annual repetition while collecting high data rates with high statistics from SORACES, we expect optimal accuracies for annual and local changes in SOR for data <1100 nm.

## 5 Conclusions

The proposed SORACES instrument is capable of recording annual changes in $SOR_a$ from 200 nm to 1100 nm, where sensitive PMT detectors provide data with a 1s cadence and a high dynamic range. Using TSI(t) with a stability of 0.01 $Wm^{-2}$ per year to normalizing TSI to ΣSSI calibrate the spectrometers and photometers, we expect accurate data close to 0.1 $Wm^{-2}$. The contribution of scattered light and optical higher orders in the spectrometer data is determined and corrected by evaluating measurements with bandpass filter s.

The proposed measurement procedure compensates for degradation through repeated calibration using the Sun as a referenced radiation source supported by solar modeling. It is integrative and does not require separate determination of optical component degradation, eliminating the influence of altitude changes and cloud coverage, uncertainties in FOV data derived from footprints, long-term radiometric drifts and other parameters.

SORACES also enables the determination of the global green Earth coverage and its annual changes by measuring
chlorophyll absorption from 350 nm to 490 nm and 620 nm to 690 nm and green backscatter from 500 nm to 600 nm. With
the high sensitivity of photometers, the mapping of the Earth would also allow to track annual local changes in green
coverage to verify the efficiency of various climate policies such as climate protection actions.

Simultaneous measurements by 12 ASSI type spectrometers with 60 PMTs and 16 photometers accumulate $1.3*10^7$ data
points for each wavelength range to be averaged for annual data or evaluated to global mapping. The high number of PMTs
and photometers offer a high level of redundancy and multiple cross-checking capability. It is possible to acquire 47 data sets
over the equator with the spectrometers and up to an additional 15 data sets per day over the polar regions from a satellite in
a sun-synchronous orbit at an altitude of 800 km during 300 days per year.

The SORACES instrument offers a variety of measurement themes by exchanging the radiation detectors and flexibly
adjusting the number of apertures.


**Acknowledgement:** We acknowledge the constructive discussions with Hauke Schmidt about climate modeling aspects,
with Margit Haberreiter and Wolfgang Finsterle about the potential of radiometers and with Gerard Thuillier. C. Jacobi
acknowledges support by Deutsche Forschungsgemeinschaft (DFG) through grant #JA836-48-1.


**Disclaimer:** The authors declare no conflict of interest.

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



Figure 1: Schematic configuration of a SORACES instrument (top view) with 12 spectrometers ASSI 1 to ASSI 12, 16 photometers P 1 to P16, and 20 radiation attenuators A 1 to A 20. ASSI spectrometers and photometers are mounted in a thermalized block below the aperture wheel. Attenuators are integrated in the aperture wheel. A cube of 50 cm$^3$ is a provisional measure of SORACES.

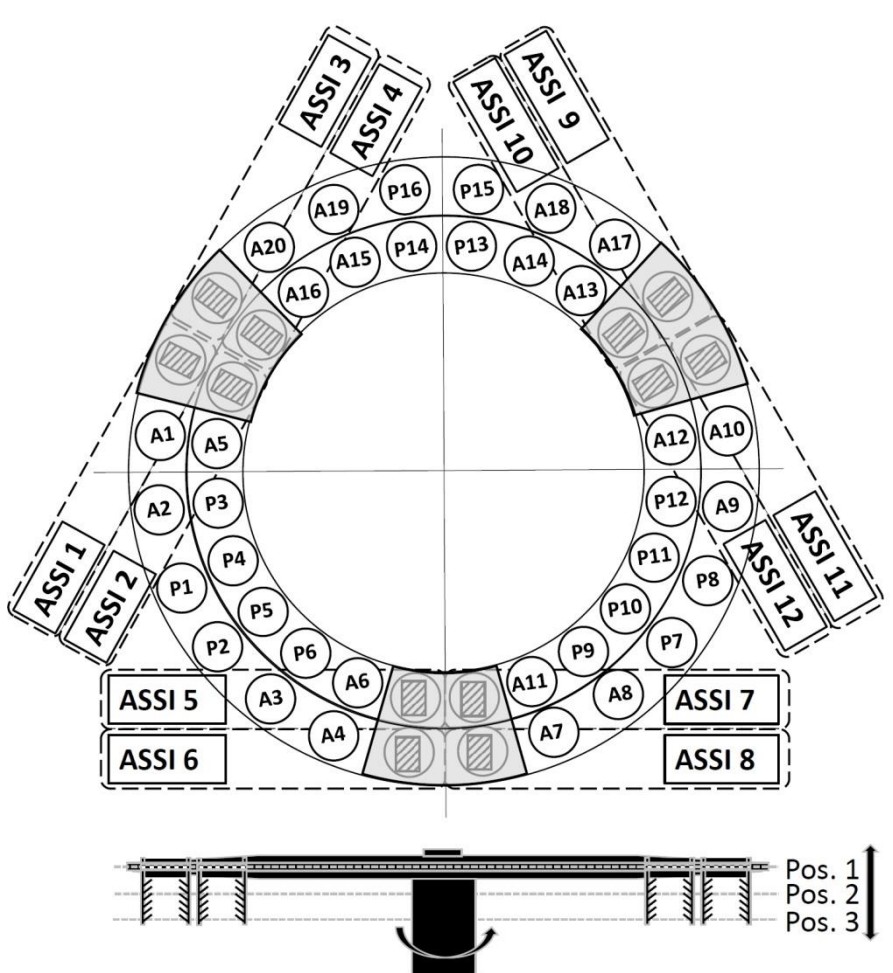



Figure 2: Absorption spectra of the chlorophyll *a* and *b* pigments in the visible light range, measured in a solvent. Both types barely absorb green light. (Milne et al., 2015).

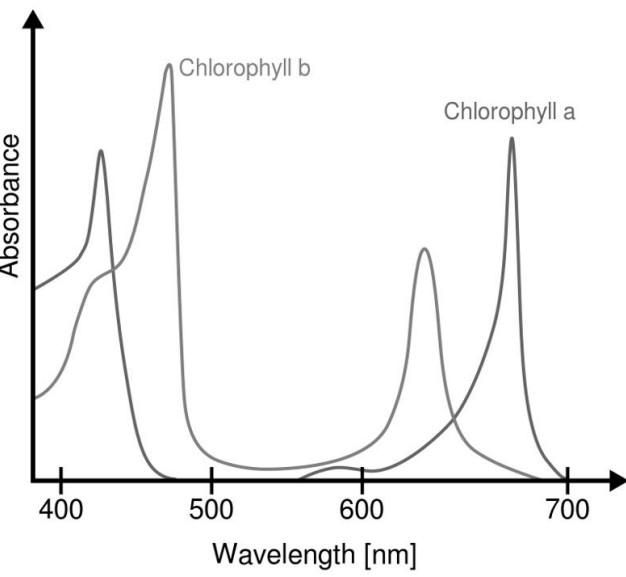


Figure 3: Spectral Outgoing Radiation enters the light path through the entrance slit to the fixed toroidal concave grating. Seven detectors behind the exits slits on the Rowland circle convert the photon fluxes in signals.

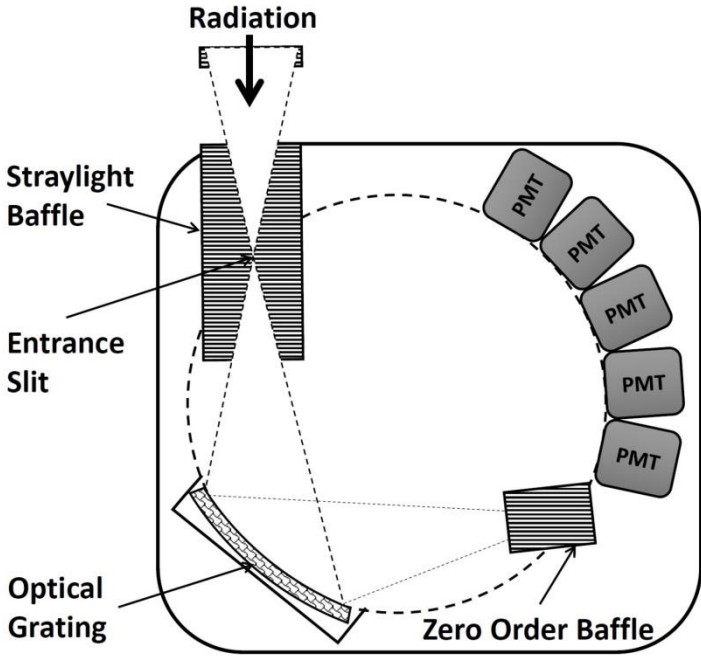