# Peer review of "Proposal of an instrument design to observe annual changes in Spectral Outgoing Radiation"

_EGUsphere, 2023_

## Referee Comment (RC2)

**Editor Review**

Dear authors,

thank you for submitting this concept paper to AMT, which has now been in the discussion stage for almost a year. After receiving the first review on 12th April, 2023, I contacted several other potential reviewers to provide a second opinion, some of whom provided feedback on aspects of the paper, but not a full review. Therefore, I now provide an editor review, summarizing external as well as my own feedback. First, I think that the concept of the paper is compelling in that it addresses the need for tracking key geophysical variables (in this case, spectral shortwave radiation from 200-1100 nm) at a stability that might enable tracking climate change – in this case, potentially over a full solar cycle. The prospect of cross-calibrating lower-accuracy sensors in orbit that the proposed high-accuracy sensor would under-fly mirrors similar ongoing efforts in the UK, the US, and elsewhere for establishing what could become a constellation of climate-observing satellite sensors in the future (in my own view). Specifically for this paper, the connection to vegetation remote sensing at a high accuracy is innovative, although it remains unclear in the manuscript why this requires an unprecedented high absolute accuracy. Usually, adequate relative accuracy is sufficient for vegetation remote sensing.

The stated goal of the paper is to describe the concept for an instrument that measures spectral outgoing radiation in a wavelength range from 200 to 1100 nm (a subset of the solar wavelength range) at a certain stability over a full solar cycle – building on a previous instrument that flew on the international space station for nine years. Stability (as opposed to accuracy) seems to be the primary focus. Clearly, the authors have deep expertise in instrumentation, and there is significant heritage for the instrument concept in general from measuring the incoming radiation, which will now be brought to bear for studying the outgoing radiation.

However, several aspects of the paper are confusing. It is unclear whether it the paper is truly meant as a proposal paper or as an initial concept paper. If it were a proposal, then specific elements would need to be included that are expected for a proposal (starting with a science question or a few science questions, deriving required observations and their attributes such as accuracy, stability, time range, spatial coverage, orbit etc., then showing that the proposed instrumentation can fulfil these requirements). If it were not a proposal but a concept paper, then it should be labeled as such so that the reader knows what to expect. As written, the direction remains unclear. The abstract seems to convey different goals than addressed in the paper later on; the introduction lists a few (valuable) science applications for the proposed technology, but there is no clear path from those to derived instrument requirements.

The confusion became apparent in the first reviewer's assessment. A few of the reviewer's questions were addressed by the authors in their responses, but several key questions remained so far unanswered, and there is no point-by-point response posted to the first reviewer's comments. It is possible that there is a misunderstanding as to what the paper entails. This was partially addressed in the authors' responses, but again, not in direct response to the reviewer's questions (point-by-point response).

The paper probably needs to be restructured, starting with the requirements given a set of science questions or objectives (this could be done with a science traceability matrix, which contains key elements such as stability, accuracy, precision), followed by an instrument description that shows how exactly those requirements are met or (if they are not met), what the path is towards fulfilling them. When rewriting the paper, it is important to not overly rely on previous publications that describe this path for a different instrument (as done currently). The new paper needs to stand on its own because (while based on a previous instrument), this is an entirely new application with different goals, requirements, and implementation strategy. The challenge will be to reconcile the various science applications that are mentioned in the introduction. For example, tracking changes in vegetation processes over time has different requirements than, for example, studying the Earth Energy Imbalance. The former requires relative accuracy of observed radiances, the latter absolute accuracy of irradiances.

As noted above, it is suggested that the authors provide (1) a clear derivation of the requirements of the mission (stability, accuracy, spectral/spatial resolution etc.) given a range of science questions or objectives, (2) describe in sufficient detail how the instrument / mission concept will be meeting them, while also considering the first reviewer's questions in this regard (comments about Line 36, 79-87, 85 etc.).

Summarizing, I would like to echo the first reviewer's recommendation to formulate the mission/instrument goals more clearly and then go the step-by-step process of a typical mission/instrument proposal.

---

## Author Comment (AC1)

**Response to RC1**

**RC1**: 'Comment on egusphere-2023-139', Anonymous Referee #1, 12 Apr 2023 reply

This paper purportedly addresses Earth's outgoing scattered radiation in the spectral range from 200-1100 nm using many instruments on a single payload to measure spectral (at some unknown resolution) and broad-band outgoing and incoming radiation. Unfortunately, the paper appears to be written rather hurriedly, making it nearly impossible to understand the full concept. In places where some aspects of the concept can be interpreted, it appears to be flawed. In order for readers to fully comprehended their instrument design, the paper needs to be rewritten in a more orderly way, and many additional details must be provided. (A few examples are in the listed comments below.)

The manuscript is rewritten. It describes a single instrument Spectral Outgoing Radiation Auto-Calibrating Spectrometers SORACES with 12 compact spectrometers and 16 photometers as sub-components, completed by 20 radiation attenuators. Each spectrometer contains 5 photomultiplier tubes (PMT), which corresponds to 60 PMT detectors covering the complete spectral range from 200 nm to 1100 nm for observing Spectral Outgoing Radiation SOR($\lambda$) with an expected stability (precision) of 0.1 W m$^{-2}$ per year.

The instrument, with simultaneously operating spectrometers and photometers, enables the determination of the global green Earth coverage and its annual changes by measuring chlorophyll absorption from 350 nm to 490 nm and 620 nm to 690 nm and green backscatter from 500 nm to 600 nm to determine a Normalized Difference Vegetation Index (NDVI). The spectral resolution between 5 nm to 40 nm is adjusted to the changes in chlorophyll absorbance with wavelength.

SORACES is not a payload, but an instrument. Spectrometers and photometes are repeatedly recalibrated with high accuracy by spectral solar irradiance in space. With a estimated dimension of 500 x 500 x 500 mm$^3$ with a weight of 46 kg, it should become part of a small climate satellite.

Another major weakness, in addition to missing fundamental information like spectral resolution, is the complete lack of measurement and mission requirements that are justified to meet science requirements. Stability is the lone variable requirement listed but the origin of that requirement is not given. Nothing is said about accuracy or precision, as if they are irrelevant.

These points have been clarified.

Another issue that need mentioning is that the measurement/instrument concept does not measure over the full shortwave spectrum nor does it measure Earth's emitted radiation. As such, it is inadequate to address the Earth's radiation imbalance. This is more of a minor comment – the authors mention energy imbalance only briefly – but considering the attention this topic is receiving in the peer-review literature, this should be recognized.

The Earth Energy Imbalance (EEI) should be corrected for the amount of solar energy stored in biomass. No determination of EEI is addressed.

One final major point: it appears that a growing trend in the community is to submit measurement and mission concepts to the peer review literature before submitting those

concepts to agencies that fund mission and instrument proposals. That is certainly fine, and in fact, welcome and encouraged, but the instrument concepts published in the peer-reviewed literature require much more detail than what is provided here. The rather ambiguous diagrams in this paper provide no insights into how a single instrument actually works, let alone several tens of instruments the authors propose. The authors even call this a "proposal" in the title. A journal paper is distinct from a proposal. (On the other hand, this does not really look like a proposal either.) I repeat what I state at the top, I think this was a hurried submission. The authors are a quite capable group of scientists. I encourage them to do a complete rewrite of this paper.

*Yes, this is not a proposal with predefined interfaces for a satellite payload. However, the method, the geometric layout and each sub-component are space-tested on rocket, satellite and ISS missions. It could be realized on the basis of this broad experience.*

Below is a only small, partial list of specific items that either need addressing or reveal fundamental flaws. Upon fully and better articulating the proposed concept, a more thorough list may be compiled.

Line 8: "From the wide range of possibilities" Awkward start to abstract. Possibilities of what?

*It was deleted.*

Line 29: I don't understand the term in parentheses in "SOR (SOR$_a$)".

*SOR$_a$ is the annual mean value of SOR(t). It has been rephrased.*

Line 32: "The proposed instrument is equipped with simultaneously measuring 12 spectrometers and 16 photometers." Perhaps: "The proposed instrument is equipped with 12 spectrometers and 16 photometers that make simultaneous measurements."

*It has been rephrased.*

Line 33: I have no idea what "20 radiation attenuators enable the adjustment of the Solar Spectral Irradiance SSI(t) to natural SOR values" means. (Edit: I think I understand after reading the entire paper that these are actually variable apertures. Those should not be called attenuators, a very misleading term.)

*It is explained in Section 4.2: The attenuators are thin metal plates with holes drilled by lasers. In this way, transmissions of $10^{-1}$, $10^{-2}$, $10^{-3}$, $10^{-4}$ and $10^{-5}$ can be obtained. The attenuators ensure a stable radiation throughput.*

Line 34: Have not identified the subscript in SOR$_a$

*It has been settled.*

Line 36: No analysis is provided to justify the adequacy the listed stability. This is, apparently a capability. What is required to meet the science goals?

*That's right: This is a capability. Stability data is not available for annual TOR data from 200 nm to 1100 nm.*

Line 54: "Besides using a different measurement method". Different from what?

It was deleted.

Lines 79-87: This entire paragraph only superficially covers how the spectrometers will be calibrated using solar irradiance. The jump from TSI to SSI is insufficient to account for spectrally varying changes in either the instruments or the Sun.

This is explained in more detail (see Introduction).

Line 85: "Normalizing TSI(t) to ΣSSI(t) adjusts the stability of both quantities so that in-space spectrometers and photometers can be calibrated with high stability to compensate for the instrument degradation." No, this is very misleading. This can only be applied uniformly across the spectrum but instrument degradation will have a wavelength dependance, as will solar variability and both of those are indistinguishable.

(see introduction): Degradation with its dependence on wavelength is ruled out by repeated calibration of the SOLACER spectrometers. After the calibration of the 8 planar grating spectrometers, the recorded SSI(t) is available as long as the instrument stability is guaranteed. 60 days per year is set to repeat calibration. If there is a significant degradation between two rounds of calibration, this can be corrected, since the degradation is usually a time-dependent process following an e-function.

Lines 88-95: This paragraph is essentially incomprehensible. It appears to use a combination of solar irradiance models and a fixed TSI measurement, neither of which is accurate enough to account for solar variability required to meet the specified stability. I trust that the authors have something else in mind but I was unable to decipher that from what was written.

(see introduction): SSI(t) should be derived from the XUV up to 3000 nm. Since TSI(t) also includes SSI(t) >3000 nm, this fraction of SSI(t) from a solar model must be added for each calibration to correctly normalize TSI(t) to ΣSSI(t).

Line 96: How does one account for the bandpass filter shapes and the fact that they will change over time?

The transmission of the bandpass filters in SolACES and SOLACER is repeatedly measured over time by placing them at the entrance of the spectrometers.

Line 113: Absorption by water vapor is curiously missing from this sentence.

Thank you, it was added.

Page 5: Much of what is on this page reads as though it is from someone's notes rather than text for a peer-reviewed manuscript.

This information serves to explain Table 1.

Line 172: "The spectral resolution should be adjusted to the requirements of the spectral regions of the observables." What are those requirements and what drives them?

The spectral resolution should be adjusted to the requirements of the spectral regions of the observables such as chlorophyll and green emission from plants.

Line 173-176: This works only if the sole source of instability is gaussian-distributed noise. There will be many other sources of instability.

$TSI_a$, $SSI_a$ and $SOR_a$ data are each a number. Multiple measurements over 300 days smooth out any sources of instability as long as there are no significant changes from one year to the next. It is a statistical method that needs to be realized and improved over time. Radiometers have taken more than 50 years to develop and are still being studied for further improvements.

Line 188: "Changes in TSI(t) cause corresponding changes in SSI(t) and SOR(t)." But the spectral compositions of those changes is unknown. Seen comment on L. 79.

Changes in the SSI(t) are known and those in the SOR(t) are measured.

Line 288: One of the co-authors (Jacobi) is listed in acknowledgements. It should be one or the other, not both.

"C. Jacobi acknowledges support by Deutsche Forschungsgemeinschaft (DFG) through grant #JA836-48-1." – is required by the DFG.

---

## Author Comment (AC2)

**RC1**: 'Comment on egusphere-2023-139', Anonymous Referee #1, 12 Apr 2023 reply - #2

What's new in the SORACES instrument?

A set of 12 spectrometers with a total of 60 photomultiplier tubes (PMTs) and 16 photometers simultaneously record SOR(t) several times per year from year to year. Each of the 76 recording channels is fully calibrated with high accuracy to compensate for degradation and/or instrumental efficiency changes. Local time is constant in a sun-synchronous orbit from a stable orbit at 800 km. SOT(t) detects radiation from clouds, atmospheric species and various surfaces 300 days resulting in an annual $SOR_a$ number from each channel. Although no absolute SOR numbers are available, there are the annual changes of interest. These directly observed $SOR_a$ data cannot be derived from oceans, which introduce average numbers over periods of up to 10 years with uncertainties due to changing ocean currents and the consideration of corresponding lower bounds of depth over the globe.

Although the SORa numbers do not represent an absolute scale, the annual changes are of interest.

The statistical annual changes in the spectral range from 200 nm to 1100 nm certainly will provide new insights in our climate system.

With my best regards,

Gerhard Schmidtke.

---

## Author Comment (AC3)

**RC1**: 'Comment on egusphere-2023-139', Anonymous Referee #1, 12 Apr 2023 reply - #3

Two points to deepen the goals of the manuscript:

1. For the relevant spectral range of 200...1100 nm, the SSI(t) data are to be determined directly with a SOLACES-type device without ionization chambers and without infrared detectors. This significantly simplifies the ongoing provision of SSI(t) for the calibration of SORACER. The estimated accuracy of 0.1 W m-2 per year is guaranteed with the reference data from the TSI instrument.

2. The absorption of chlorophyll and the reflection from the green part in the plants are very pronounced and can each be measured with high signal-to-noise ratio, as shown in the following representative example. SSI(t) data are of the mean order of $7.5*10^{13}$ photons m$^{-2}$ s$^{-1}$ entering the ASSI entrance slit. Using the data of Figure 16 below, $5.3*10^7$ cps would characterize the green and $2.7*10^7$ cps the chlorophyll ASSI signals. By attenuating the TSI(t) by three orders of magnistude, the full dynamic range of ASSI at almost seven orders of magnitude would be available for instrument calibration and observation of global vegetation. – The green ánd chlorophyll dada are in good agreement with other publications (see below).

**Detection of sugarbeet diseases using remote sensing techniques**

[Figure]

SUGAR–BEETS

Fig. 16

- Offentliggjort: 1988
- Forfatter(e): Europa-Kommissionen
- https://op.europa.eu/da/publication-detail/-/publication/d4a320c7-3ad9-4e7e-b763-925f81fa2df0

******************************

**Chlorophyll does not reflect green light – how to correct a misconception**

Olli Virtanena Department of Biochemistry/Molecular Plant Biology, University of Turku, Turku, Finland    https://orcid.org/0000-0002-2991-520X

[Figure]

[Figure]

Reflectance spectra of green (black, solid line) and yellow (red, dashed line) leaves of B. pendula (a) and examples of the leaves (b). Specular reflectance was measured with an STS-VIS spectrometer, using a 250 W halogen lamp as a light source. For the measurement, a leaf disk was placed on a matt black cardboard at a 5 mm distance from the probe, and the probe was aligned with the surface normal. Each curve represents an average of 6 independent biological replicates, and the data have been smoothened with a moving median using a window of 9 data points.

************************************

Commission VII , W. G. VII / 9

H .- J . BOEHNEL , W. FISCH£R,G. KNOLL,

Fraunhofer - Institut fur Physikalische Messtechnik (former Institut fur Physikalische Weltraumforschung) Freiburg i . Br ., Federal Republic of Germany

THE DEPENDENCE OF THE SPECTRAL SIGNATURE OF SUGARBEETS ON THE OBSERVATION LEVEL AND THE REFLECTION GEOMETRY .

https://www.isprs.org/proceedings/XXIII/congress/part7-8/102_XXIII-B7-8.pdf

******************************

With my best regards,

Gerhard Schmidtke.

---

## Author Comment (AC4)

**Response #2 to RC1**

Thank you for the insightful comments, which will very much help us to substantially improve the paper. Please find below our responses to the reviewers remarks. We will repeat these in italics and add our respective responses in normal letters.

*This paper purportedly addresses Earth's outgoing scattered radiation in the spectral range from 200-1100 nm using many instruments on a single payload to measure spectral (at some unknown resolution) and broad-band outgoing and incoming radiation. Unfortunately, the paper appears to be written rather hurriedly, making it nearly impossible to understand the full concept. In places where some aspects of the concept can be interpreted, it appears to be flawed. In order for readers to fully comprehended their instrument design, the paper needs to be rewritten in a more orderly way, and many additional details must be provided. (A few examples are in the listed comments below.)*

Response: Thank you for your comment. We will slightly modify the title of the paper:

> "An instrument design to observe annual changes in Spectral Outgoing Radiation from 200-1100 nm"

The proposed instrument design is called Spectral Outgoing Radiation Auto-Calibrating Spectrometers SORACES and is intended for the long-term measurement of annual changes in spectral outgoing radiation SOR($\lambda$,t) from 200-1100 nm. Two topics are targeted by remote sensing during a solar cycle period: the determination of annual changes in global vegetation cover in the spectral range 350 nm to 700 nm and spectral outgoing radiation SOR from 200 nm to 1100 nm. A global mapping of the vegetation area and the SOR as well as their annual changes should show the results.

(Note: In order to reinforce the topics of annual changes in vegetation cover and SOR, we have excluded the exploration of local changes in vegetation cover due to the complexity of the subtopic. This is achieved by using additional ASSI-type spectrometers instead of the photometers. It also improves redundancy and internal cross-calibration, which are important aspects of long-term observations.)

Based on the actual knowledge of the total solar irradiance TSI(t) from radiometers (Schmutz et al., 2013; Scafetta and Willson, 2014) and the spectral solar irradiance SSI($\lambda$,t) from a Solar Auto-Calibrating XUV-IR Spectrometer instrument SOLACER (Schmidtke et al. 2019), SORACES applies the measurement method of the Solar Auto-Calibrating EUV Spectrometers SolACES (Schmidtke et al., 2014; Schaefer et al., 2017). Using 16 Airglow Solar Spectrometer Instruments ASSI (Schmidtke et al., 1985) with a total of 80 channels operating simultaneously, spectral outgoing radiation SOR($\lambda$,t) of 200-1100 nm shall be observed with a higher spectral resolution of 350-700 nm. A schematic of the arrangement of the spectrometers is added at the end of this response.

To reduce the influence of drag on the accuracy of the data and the SOR dependence on local time, an orbital altitude of 800 km in a Sun-synchronous orbit is chosen. Lower orbits such as the ERBS orbit result in irregular solar activity-dependent altitude changes with decreasing target area of SOR data of 7 % (see manuscript Lines 233-240). The constant fields of view of all sub-instruments are 15 by 30 degrees. In this way, the annual signal averages of each detector provide SOR information on annual changes in the same target area.

Operational management, data handling and data evaluation are supported by artificial intelligence (AI) algorithms to be developed on ground and adapted to the measurement conditions during the mission. Pre-calibration in the laboratory and using the Sun enable the handling of the 80 detectors. The application of Lambert screen rules is used to provide the AI

data pool from various green scenes such as grasslands or forests to be observed from rooftops, observation towers, drones and/or aircraft.

*Another major weakness, in addition to missing fundamental information like spectral resolution, is the complete lack of measurement and mission requirements that are justified to meet science requirements. Stability is the lone variable requirement listed but the origin of that requirement is not given. Nothing is said about accuracy or precision, as if they are irrelevant.*

Response: Thank you for this point. We will make this clearer to the reader in the revised version

Requirements - accuracy, precision/stability, spectral/spatial resolution:

Since neither annual changes in SOR data from 200-1100 nm nor in chlorophyll from 350-700 nm are known, we must use available instruments and data instead of formulating requirements. In this context we choice TSI(t) data as the main input with the given accuracy and the precision/stability of the radiometers. To determine the annual changes in SOR, the long-term stability of the measured values is the most important parameter, which is 5-7 mW $m^{-2}$ $yr^{-1}$ (Montillet et al., 2022). We call the long-term stability the annual average of precision. The accuracy of mW m-2 is not of interest in this context.

In order to repeatedly calibrate the ASSI spectrometers with high long-term stability to compensate for the degradation, we normalize

$$\int SSI(\lambda, t)d\lambda = TSI(t) \hspace{3cm} (1).$$

In this way we should achieve stability of 0.1 $Wm^{-2}$ over a period of a solar cycle or longer.

A radiometer for TSI(t) data and a SOLACER instrument for SSI($\lambda, t$) data could be operated aboard the same or a different spacecraft.

Vegetation observation should be carried out by evaluating data in the spectral range from 350-700 nm due to absorption by chlorophyll and by scattering of green radiation. The chlorophyll absorption bands extend from 350 nm to 490 nm and from 620 nm to 690 nm. The green backscattering is active from 500 nm to 600 nm. In these regions, the spectral resolution changes from 5 nm to 40 nm depending on the SOR backscatter curvature.

To observe annual SOR changes from 200-1100 nm, the spectral resolution ranges from 20-50 nm outside of the range 350-700 nm.

The field of view is another parameter. Each sub-instrument is designed to monitor the same areas of 15 x 30 degrees.

*Another issue that needs mentioning is that the measurement/instrument concept does not measure over the full shortwave spectrum nor does it measure Earth's emitted radiation. As such, it is inadequate to address the Earth's radiation imbalance. This is more of a minor comment – the authors mention energy imbalance only briefly – but considering the attention this topic is receiving in the peer-review literature, this should be recognized.*

Response: Measuring of the Earth Energy Imbalance is not addressed in the manuscript. Instead, the need to include the amount of absorbed solar energy resulting in biomass should be considered in terms of a correction factor in Earth's energy imbalance (see manuscript Lines 2, 45/46 and 54/55). Since this aspect is not relevant to the SOR measurements, it will be deleted.

*One final major point: it appears that a growing trend in the community is to submit measurement and mission concepts to the peer review literature before submitting those concepts to agencies that fund mission and instrument proposals. That is certainly fine, and in*

*fact, welcome and encouraged, but the instrument concepts published in the peer-reviewed literature require much more detail than what is provided here. The rather ambiguous diagrams in this paper provide no insights into how a single instrument actually works, let alone several tens of instruments the authors propose. The authors even call this a "proposal" in the title. A journal paper is distinct from a proposal. (On the other hand, this does not really look like a proposal either.) I repeat what I state at the top, I think this was a hurried submission. The authors are a quite capable group of scientists. I encourage them to do a complete rewrite of this paper.*

Response: Yes, the paper will be completely rewritten. It is not a proposal with predefined interfaces for a satellite payload. However, for the reader's better understanding, the method, geometric structure and each sub-instrument should be described. They are tested in space on rocket, satellite and ISS missions. SORACES could be realized on the basis of this broad experience.

*Below is a only small, partial list of specific items that either need addressing or reveal fundamental flaws. Upon fully and better articulating the proposed concept, a more thorough list may be compiled.*

Response: Thank you for this list. Below we will respond to the single items.

*Line 8: "From the wide range of possibilities" Awkward start to abstract. Possibilities of what?*

Response: This will be deleted.

*Line 29: I don't understand the term in parentheses in "SOR ($SOR_a$)".*

Response: $SOR_a$ is the annual mean value of SOR(t). It will be reworded.

*Line 32: "The proposed instrument is equipped with simultaneously measuring 12 spectrometers and 16 photometers." Perhaps: "The proposed instrument is equipped with 12 spectrometers and 16 photometers that make simultaneous measurements."*

Response: This will be rephrased.The photometers are deleted.

*Line 33: I have no idea what "20 radiation attenuators enable the adjustment of the Solar Spectral Irradiance SSI(t) to natural SOR values" means. (Edit: I think I understand after reading the entire paper that these are actually variable apertures. Those should not be called attenuators, a very misleading term.)*

Response: This is explained in Section 4.2: The attenuators are thin metal plates with holes drilled by lasers. In this way, transmissions of $10^{-1}$, $10^{-2}$, $10^{-3}$, $10^{-4}$ and $10^{-5}$ can be obtained. The attenuators ensure a stable radiation throughput. They attenuate or weaken the TSI to the optimum levels of the sub-instrumental data statistics. 32 attenuators can be placed in the attenuator wheel (see schematic view below) to compensate for different spectrometer efficiencies. They can also be placed in front of the detectors in the spectrometer. In this way, optimal count rates are achieved for each of the 80 detectors. This will be explained in detail.

*Line 34: Have not identified the subscript in $SOR_a$*

Response: This is explained in the comment on line 29.

*Line 36: No analysis is provided to justify the adequacy the listed stability. This is, apparently a capability. What is required to meet the science goals?*

Response: This is explained on pages 1 and 2 of this response: To meet the science goals, a stability of 0.1 Wm$^{-2}$ over a period of a solar cycle is required.

*Line 54: "Besides using a different measurement method". Different from what?*

Response: This will be deleted.

*Lines 79-87: This entire paragraph only superficially covers how the spectrometers will be calibrated using solar irradiance. The jump from TSI to SSI is insufficient to account for spectrally varying changes in either the instruments or the Sun.*

Response: A reference to an earlier publication replaces this explanation.

*Line 85: "Normalizing TSI(t) to ΣSSI(t) adjusts the stability of both quantities so that in-space spectrometers and photometers can be calibrated with high stability to compensate for the instrument degradation." No, this is very misleading. This can only be applied uniformly across the spectrum but instrument degradation will have a wavelength dependance, as will solar variability and both of those are indistinguishable.*Response:

Respone: Using the SSI(t) data from a Solar Auto-Calibrating XUV-IR Spectrometer System SOLACER (Schmidtke et al., 2919), degradation with its dependence on wavelength is excluded by repeated calibration of the SORACER sub-instruments with the known spectral distribution SSI(t)  in the range from 200-1100 nm. Then the individual photon effciency of each sub-instrument is known over time. In this way, the time and wavelength dependence are taken into account. 60 days per year is set to repeat the calibration. If there is significant degradation between two rounds of calibration, it can be corrected, since degradation is usually a time-dependent process following an e-function.

*Lines 88-95: This paragraph is essentially incomprehensible. It appears to use a combination of solar irradiance models and a fixed TSI measurement, neither of which is accurate enough to account for solar variability required to meet the specified stability. I trust that the authors have something else in mind but I was unable to decipher that from what was written.*

Response: See answer to Line 85 and Eq. 1.

*Line 96: How does one account for the bandpass filter shapes and the fact that they will change over time?*

Response: No bandpass filters are used in SORACER.

*Line 113: Absorption by water vapor is curiously missing from this sentence.*

Response: This will be added.

*Page 5: Much of what is on this page reads as though it is from someone's notes rather than text for a peer-reviewed manuscript.*

Response: This information was used to explain Table 1. It will be shortened.

*Line 172: "The spectral resolution should be adjusted to the requirements of the spectral regions of the observables." What are those requirements and what drives them?*

Response: See Page 2 below Eq. 1: Spectral resolution is adjusted to the curvature of the SOR. Significant changes in curvature occur in the spectral range of the chlorophyll

absorption bands, which requires a spectral resolution of up to 5 nm, whereas a flat curvature requires a spectral resolution of up to 50 nm

*Line 173-176: This works only if the sole source of instability is gaussian-distributed noise. There will be many other sources of instability.*

Response: Annual $TSI_a$, $SSI_a$ and $SOR_a$ data are each a number. Multiple measurements over 300 days smooth out any sources of instability. It is a statistical method based on collecting data according to the same pattern year after year. The stability of the $TSI_a$ data as a reference includes all sources of instability.

*Line 188: "Changes in TSI(t) cause corresponding changes in SSI(t) and SOR(t)." But the spectral compositions of those changes is unknown. Seen comment on L. 79.*

Response: Spectral composition changes in the $SSI(\lambda,t)$ are known input from SOLACER.

*Line 288: One of the co-authors (Jacobi) is listed in acknowledgements. It should be one or the other, not both.*

Response: "C. Jacobi acknowledges support by Deutsche Forschungsgemeinschaft (DFG) through grant #JA836-48-1." – is required by the DFG.

References

Montillet, J.-P., W. Finsterle, G. Kermarrec, R. Sikonja, M. Haberreiter, W. Schmutz, and T. Dudok de Wit: Data Fusion of Total Solar Irradiance Composite Time Series Using 41 Years of Satellite Measurements, J. Geophys. Res..: Atmospheres 127(13), 2022, https://doi.org/10.1029/2021JD036146.

Schaefer, R., Schmidtke, G., Strahl, T., Pfeifer, M., and Brunner, R.: EUV data processing methods of the Solar Auto-Calibrating EUV Spectrometers (SolACES) aboard the International Space Station, Adv. Space Res., 59, 2207-2228, DOI: 10.1016/j.asr.2017.02.036, 2017.

Schmidtke, G., Seidl P., and Wita, C.: Airglow-solar spectrometer instrument 20-700 nm aboard the San Marco D/L satellite, Appl. Optics 24, 3206-3213, doi: 10.1364/ao.24.003206, 1985.

Schmidtke, G., Nikutowski, B., Jacobi, Ch., Brunner, R., Erhardt, Ch., Knecht, S., Scherle, J., and Schlagenhauf, J.: Solar EUV irradiance measurements by the Auto-Calibrating EUV Spectrometers (SolACES) aboard the International Space Station (ISS), Solar Phys., 289, 1863-1883, DOI: 10.1007/s11207-013-0430-5, 2014.

Schmidtke, G., Finsterle, W., van Ruymbeke, M., Haberreiter, M., Schäfer, R., Zhu, P., and Brunner, R.: Solar Auto-Calibrating XUV-IR Spectrometer System (SOLACER) for the measurement of solar spectral irradiance, Appl. Opt., **58**(22), 6182-6192 , doi.org/10.1364/AO.58.006182, 2019.

NEU April 2024

SOR 01.jpg

[Figure]

Schematic of the spectromters: the attenuator wheel

---

## Author Comment (AC5)

**Response to RC2**

Thank you for the editor's review, which will greatly help us to improve the manuscript significantly. Please find below our responses to the editor's remarks. We will repeat these in italics and add our respective responses in normal letters.

*Dear authors,*

*thank you for submitting this concept paper to AMT, which has now been in the discussion stage for almost a year. After receiving the first review on 12th April, 2023, I contacted several other potential reviewers to provide a second opinion, some of whom provided feedback on aspects of the paper, but not a full review. Therefore, I now provide an editor review, summarizing external as well as my own feedback. First, I think that the concept of the paper is compelling in that it addresses the need for tracking key geophysical variables (in this case, spectral shortwave radiation from 200-1100 nm) at a stability that might enable tracking climate change – in this case, potentially over a full solar cycle. The prospect of cross-calibrating lower-accuracy sensors in orbit that the proposed high-accuracy sensor would under-fly mirrors similar ongoing efforts in the UK, the US, and elsewhere for establishing what could become a constellation of climate-observing satellite sensors in the future (in my own view). Specifically for this paper, the connection to vegetation remote sensing at a high accuracy is innovative, although it remains unclear in the manuscript why this requires an unprecedented high absolute accuracy. Usually, adequate relative accuracy is sufficient for vegetation remote sensing.*

Thank you for this summary. We will respond to your and other reviewers' criticisms in a revised version in order to describe the concept of the instrument more clearly.

*The stated goal of the paper is to describe the concept for an instrument that measures spectral outgoing radiation in a wavelength range from 200 to 1100 nm (a subset of the solar wavelength range) at a certain stability over a full solar cycle –building on a previous instrument that flew on the international space station for nine years. Stability (as opposed to accuracy) seems to be the primary focus. Clearly, the authors have deep expertise in instrumentation, and there is significant heritage for the instrument concept in general from measuring the incoming radiation, which will now be brought to bear for studying the outgoing radiation. However, several aspects of the paper are confusing. It is unclear whether it the paper is truly meant as a proposal paper or as an initial concept paper. If it were a proposal, then specific elements would need to be included that are expected for a proposal (starting with a science question or a few science questions, deriving required observations and their attributes such as accuracy, stability, time range, spatial coverage, orbit etc., then showing that the proposed instrumentation can fulfil these requirements). If it were not a proposal but a concept paper, then it should be labelled as such so that the reader knows what to expect. As written, the direction remains unclear. The abstract seems to convey different goals than addressed in the paper later on; the introduction lists a few (valuable) science applications for the proposed technology, but there is no clear path from those to derived instrument requirements.*

Thank you for pointing this out. In the revised version we will make the purpose of the paper more clear.

*The confusion became apparent in the first reviewer's assessment. A few of the reviewer's questions were addressed by the authors in their responses, but several key questions remained so far unanswered, and there is no point-by-point response posted to the first reviewer's comments. It is possible that there is a misunderstanding as to what the paper entails. This was partially addressed in the authors' responses, but again, not in direct response to the reviewer's questions (point-by-point response).*

We have also submitted a point-to-point response to the reviewers´ comments now.

*The paper probably needs to be restructured, starting with the requirements given a set of science questions or objectives (this could be done with a science traceability matrix, which contains key elements such as stability, accuracy, precision), followed by an instrument description that shows how exactly those requirements are met or (if they are not met), what the path is towards fulfilling them. When rewriting the paper, it is important to not overly rely on previous publications that describe this path for a different instrument (as done currently). The new paper needs to stand on its own because (while based on a previous instrument), this is an entirely new application with different goals, requirements, and implementation strategy. The challenge will be to reconcile the various science applications that are mentioned in the introduction. For example, tracking changes in vegetation processes over time has different requirements than, for example, studying the Earth Energy Imbalance. The former requires relative accuracy of observed radiances, the latter absolute accuracy of irradiances.*

Thank you. We will completely restructure and modify the paper.

*As noted above, it is suggested that the authors provide (1) a clear derivation of the requirements of the mission (stability, accuracy, spectral/spatial resolution etc.) given a range of science questions or objectives, (2) describe in sufficient detail how the instrument / mission concept will be meeting them, while also considering the first reviewer's questions in this regard (comments about Line 36, 79-87, 85 etc.).*

*Summarizing, I would like to echo the first reviewer's recommendation to formulate the mission/instrument goals more clearly and then go the step-by-step process of a typical mission/instrument proposal.*

Thank you. We will rewrite the paper according to your and the reviewer´s suggestions.

---

## Author Comment (AC6)

**6 Conclusions and the need to create referenced TSI and SSI indices**

"A plausible simulation of the global energy balance is a first-order requirement for a credible climate model.", (Wild, 2020). Since the determination of EEI requires an accuracy of $\approx$0.1 W m$^{-2}$ or 7*10^-3 %, TSI, TOR and SOR instruments must meet this high performance in space and be calibrated correspondingly. This assumes the availability of simulating TSI and TOR radiation sources with an accuracy of 0.1 W m$-2$, which is not possible. Neither the Sun, with a spectral composition of regions with 6000 K to a maximum of 200 million K, nor the Earth, with temperatures from -80 K to +60 K and reflected solar radiation in the VUV to infrared spectral range, can be simulated in the laboratory or in space. If the effects of permanent degradation are also taken into account, the dilemma of the impossibility of determining the EEI with the required accuracy becomes more understandable.

A reasonable approach is to derive annual changes in SOR data from SORACES in the spectral region from 200 to 1100 nm. Using data from DARA instruments and deriving reference SSI(t) data from a SOLACER instrument, a stability close to 5 mW m$^{-2}$ year$^{-1}$ over a solar cycle or more is expected. With regularly updating the efficiency of the SORACES spectrometers, the degradation will be compensated. 80 PMT channels provide TOR spectra with a cadence of 1 s$^{-1}$. The instrument also enables the determination of the global green Earth coverage and its annual changes by measuring chlorophyll absorption from 350 nm to 490 nm and 620 nm to 690 nm and green backscatter from 500 nm to 600 nm. With the high sensitivity, the mapping of the Earth would also allow to track annual local changes in green coverage to verify the effectiveness of various climate policies such as climate protection actions. With the numerous repeated measurements of the areas over a period of the year, similar influences of the respective cloud cover are assumed. They can be recorded by evaluating the measured with selected model spectra. With a rigorous operational schedule for collecting high data rates with high statistics from SORACES, we expect optimal accuracies for annual and local changes in SOR(t) for data <1100 nm.

A great advantage of this development is the ability to test the expected SOR data (Table 1) from the ground and from balloons, collect the AI data pool before launch and create reference SSI(t) data also from ground measurements. It is intended to use these data for SSI(t)-related instruments in space and for climate modeling.

In principle, a similar instrument TORACES should be investigated by extending the wavelength range >1100nm. For example, increasing the detection area of BOS sensors (Zhu et al., 2015) from 1 cm to 5 cm diameter could provide a factor of 25 to increase sensitivity. A stable Black Body radiator at a medium Earth temperature in space should be developed for as radiation reference radiation source and contribute to further improvements in space. While SORACES could provide shortwave TOR-SW data, TORACES with the reference radiation source should provide longwave data TOR-LW. However, implementing such a tool, important as it is for climatology, is still to be investigated.

It is proposed to operate two SOLACER instruments, in space and on the ground, to simultaneously derive SSI(t) data. Ultimately, a SOLACER instrument with its two radiometers should be able to determine permanent SSI(t) reference data from the ground and correct the atmospheric absorption by comparison with the SSI(t) data in orbit.

Our approach for creating TSI and SSI indices like the reference meter in Paris is considered an important requirement for future research in climatology. It suggests the development of a climate satellite with the following payload: A TSI and a TOR instrument, SOLACER, SORACES, instruments from CERES, ERBE, CLARREO and/ other programs.